# Old and New Biomarkers for Infection, Inflammation, and Autoimmunity in Treatment-Resistant Affective and Schizophrenic Spectrum Disorders

**DOI:** 10.3390/ph15030299

**Published:** 2022-02-28

**Authors:** Christian Scheiber, Tanja Schulz, Julian M. Schneider, Karl Bechter, E. Marion Schneider

**Affiliations:** 1Division of Experimental Anaesthesiology, Ulm University Hospital, 89081 Ulm, Germany; christian-1.scheiber@uni-ulm.de (C.S.); tanja.schulz@uni-ulm.de (T.S.); julian.schneider@uniklinik-ulm.de (J.M.S.); 2Clinic for Psychiatry and Psychotherapy II, Ulm University, 89312 Günzburg, Germany; karl.bechter@bkh-guenzburg.de

**Keywords:** affective disorders, schizophrenia, cerebrospinal fluid, biomarkers, immune phenotypes, monocytes

## Abstract

Affective (AF) and Schizophrenic (SZ) Spectrum disorders manifest with risk factors, involving inflammatory processes linked to infections and autoimmunity. This study searched for novel biomarkers in cerebrospinal fluid (CSF) and peripheral blood. A total of 29 AF and 39 SZ patients with treatment-resistant disease were included. In CSF, the chemokine IL-8 was significantly elevated in AF and SZ patients. IL-8 promotes chemotaxis by neutrophils and may originate from different tissues. S100B, a glia-derived brain damage marker, was higher in CSF from AF than SZ patients. Among the plasma-derived biomarkers, ferritin was elevated in AF and SZ. Soluble CD25, indicating T_reg_ dysfunction, was higher in SZ than in AF patients. Interferon-γ, implying virus-specific immune activation, was positive in selective AF patients, only. Both groups showed elevated expression of immunosuppressive CD33 on monocytes, but higher amounts of CD123^+^ plasmacytoid dendritic cells were restricted to SZ. In conclusion, chemotactic IL-8 indicates neuronal stress and inflammation in the CSF of both groups. Novel plasma-derived biomarkers such as sCD25 and monocytic CD33 distinguish SZ from AF with an autoimmune phenotype.

## 1. Introduction

Psychiatric diseases, including Affective (AF, ICD-10 F30-F33) and Schizophrenic (SZ, ICD-10 F20-F25) Spectrum disorders, represent a group of detrimental diseases with their etiology not yet completely understood. Besides social and genetic risk factors [1,2], a growing body of evidence supports inflammation as a key player in their pathogenesis [3,4]. Psychiatric diseases may also arise from metabolic inflammation, pathogen-induced inflammation, or sterile inflammation from tissues damage [5]. Immune profiles, indicating inflammation and infections can be identified by biomarkers. As previously demonstrated [6,7], biomarkers may indicate oxidative stress, infections, and immune activation. Plasma cytokines are often macrophage-derived cytokines such as interleukin-(IL)-1β, IL-6, Tumor necrosis factor (TNF)-α, and IL-8, but do not always mirror the cytokine pattern found in cerebrospinal fluid [5]. Insufficiency of regulatory T cells (T_reg_) [8] could influence the amount of circulating CD25, the soluble alpha chain of the high affinity IL-2 receptor which is also related to T-cell and NK-cell activation in general [9]. IL-10 is an immunosuppressive cytokine produced by monocytes and glia cells, as well as EBV-infected B-cells [10], and type 1 regulatory (T_r1_) T cells [11]. This cytokine has been proven to be related to schizophrenia in both animal and human models [12]. IL-10 is also implicated in depression, especially in untreated patients, as recently reviewed [13]. S100B represents the so-called type 1 calcium-binding protein B (S100B), a marker of neuronal cell damage [5,14]. To address the potential inflammation caused by virus infections or virus reactivation, interferons constitute valuable biomarkers. Viruses lead to T- and natural killer (NK) cell activation and type II interferon (IFN-γ) release [15]. Moreover, nucleic acids, derived from either tissue damage or viruses themselves, however, would result in increased type I interferons [16]. Finally, certain biomarker patterns indicate sterile inflammation, e.g., when infection-associated biomarkers (such as IL-6) are low, but IL-1β is elevated. This condition is expected to result from NLRP3 inflammasome activation [17]. A more advanced approach uses cellular immune profiles determined by flow cytometric analysis of white blood cells using antibodies against clusters of differentiation (CD) antigens, receptors, transcription factors, as well as cytoplasmic cytokines and functional vesicles such as perforin [18]. We here included immune phenotypic data from patients with AF and SZ to better study the inflammatory pathway and functional impact of novel markers such as CD11b and CD33 in AF and SZ impacting autoimmune dysfunction.

## 2. Results

### 2.1. Plasma and CSF Cytokine Profiles

Peripheral blood plasma and cerebrospinal fluid (CSF) were tested for pro-inflammatory biomarkers IL-8 and IL-1β, as well as the anti-inflammatory marker IL-10. Figure 1 shows that median values of IL-8 concentrations were significantly higher in CSF as compared to plasma of both patient groups, AF (*p* = 0.002, Figure 1A) and SZ (*p* < 0.0001, Figure 1A). However, CSF IL-8 levels from both patient groups were low when compared with CSF obtained from patients with Subarachnoid hemorrhage (SAH) (Appendix A). Median IL-10 concentrations were also higher in CSF of AF patients (*p* = 0.02, Figure 1B), but not in SZ. Median values of IL-1β were higher in plasma samples of SZ patients when compared to CSF (*p* = 0.02, Figure 1C). The IL-8 results provide evidence for significant inflammation and oxidative stress in the CSF of both patient groups which is in part balanced by anti-inflammatory IL-10. However, despite similar plasma levels, inflammasome-related IL-1β is more prevalent in the peripheral blood of SZ rather than AF patients.

Cytokine measurements also included IL-6. Median plasma concentrations were not different, but individual patients of the SZ cohort presented with highly elevated plasma IL-6 as well as somewhat elevated IL-6 in CSF (Figure 1D). Apart from some outliers, TNF-α was not different between patients’ groups of both plasma and CSF. For IL-8, IL-10, IL-1β, IL-6, and TNF-α, plasma concentrations of both patients’ groups were not significantly different from HDs. Due to its relevance as a biomarker for traumatic stress, we also tested S100B in CSF of both patient groups, and CSF samples isolated from SAH patients were included as well. Due to traumatic brain injury in SAH, S100B concentrations were highly elevated in SAH patients (median: 768.70 pg/mL), followed by AF patients (median: 191.00 pg/mL, *p* = 0.18) and SZ patients (median: 93.87 pg/mL, *p* = 0.02) (Figure 1F). Another plasma marker, lipopolysaccharide-binding protein (LBP), indicating immune activation against circulating endotoxin, was not different between patients and HDs (Appendix A). Additionally, we quantified type I interferon (IFN-α) and found no evidence for significantly different concentrations (Appendix A). IFN-α was below the detection level of our ELISA (<10 pg/mL), except for a single plasma sample in the SZ group.

### 2.2. Immune Phenotype Stratification

To further substantiate inflammatory and infectious complications in psychiatric patients, leukocyte subpopulations were characterized and compared with HDs. We used flow cytometry to calculate the relative amounts of lymphocytes, monocytes, and neutrophilic granulocytes as well as expression densities by defined CD. Figure 2 shows that there was a trend for higher lymphocyte counts in AF, and significantly higher monocyte counts in SZ compared to HDs (*p* = 0.01). Major differences between patients’ groups and HDs were not detected We also calculated the neutrophil–lymphocyte ratio (LNR) and the lymphocyte–monocyte (LMR) ratio (Table 1).

Median values of LMRs were significantly lower in SZ when compared with HDs. Interestingly, the two SZ patients with the highest monocyte counts had the lowest relative amounts of neutrophilic granulocytes (Figure 2B,C red and blue symbols).

Immune phenotypes using CD-defined antigens were used to identify profiles related to: (i) inflammation (Figure 3), (ii) infection (Figure 4), and (iii) autoimmunity (Figure 5).

### 2.3. Antigen Presenting Cells (APCs)

Antigen-presenting cells (APCs), comprised of monocytes and dendritic cells (DC), are the primary leukocyte population indicating, and also triggering inflammation. For subclassification of inflammatory (M1) vs. anti-inflammatory (M2) monocytes, we used CD163. As demonstrated in Figure 3A, AF and SZ patients had less CD163^+^ (M2) monocytes than the majority of HDs. The expression of the plasmacytoid dendritic cell (pDC) marker CD123 was higher in SZ than in AF and HDs (Figure 3B). The expression densities of the immunosuppressive ligand CD33 were tested to evaluate the differentiation state of blood-derived monocytes since CD33 is highly upregulated in immature monocytes. Figure 3C shows that monocytes of both AF and SZ had higher CD33 expression densities than HDs, with the effect being more prominent in SZ than in AF. To judge the capacity of migration by leukocytes, we quantified CD11b, the ligand for ICAM-1 (intercellular adhesion molecule 1, also named CD54), which is expressed by endothelial cells. Results on CD11b expression densities on monocytes are shown in Figure 3D. With a few exceptions in both patient groups, CD11b expression was not significantly different from healthy HDs. HLA-DR, MHC-class II antigens, are highly relevant to evaluate the immune competence of APCs. As shown in Figure 3E, individual patients of the AF and also the SZ group expressed very high levels of HLA-DR; however, median values of the cohorts did not differ from HDs. In summary, the M2 marker CD163 and the immaturity marker CD33 were remarkably different between patients and non-affected individuals. In order to include a marker for the activation of M1 macrophages we studied the plasma concentrations of ferritin (Figure 3F). Concentrations were higher in both AF (*p* = 0.02) and SZ patients (*p* = 0.04), when compared to HDs. Differences between AF and SZ patients were not significant. Figure 3F further shows that differences in the median values between AF and SZ patients correspond to M2 (CD163) and immaturity markers (CD33).

**Figure 3 pharmaceuticals-15-00299-f003:**
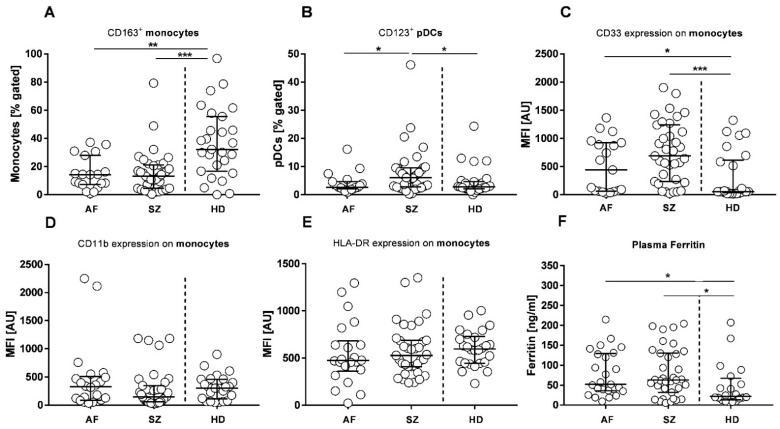
Flow cytometric profiles of antigen-presenting cells (APCs) in patients with AF, SZ, and HDs. Relative amounts (% gated) of CD163^+^ leukocytes (AF/HDs: median: 14.23 vs. 32.22, *p* = 0.001; SZ/HDs: median: 13.11 vs. 32.22, *p* ≤ 0.001) (**A**) and CD123^+^ pDCs (median: 6.060 vs. 2.650, *p* = 0.02 (**B**) are shown. Expression densities of CD33 (AF/HDs: median: 441.0 vs. 56.24, *p* = 0.03; SZ/HDs: median: 691.2 vs. 56.24, *p* ≤ 0.001) (**C**), CD11b (**D**) and HLA-DR (**E**) are shown and given as arbitrary units (AU). Ferritin plasma concentrations: (AF/HDs: median: 52.00 ng/mL vs. 22.20 ng/mL, *p* = 0.02; SZ/HDs: median: 62.80 ng/mL vs. 22.20 ng/mL, *p* = 0.04) are shown in (**F**). All results are displayed as scatter plots, with bars indicating the median including interquartile ranges. Wilcoxon rank-sum test was performed to analyze differences among groups, with significant differences being marked by * *p* ≤ 0.05, ** *p* ≤ 0.01, and *** *p* ≤ 0.001.

### 2.4. Pathogen Receptors (Toll-like Receptors, TLRs)

Further analysis addressed the modulation of pathogen receptors TLR2 and TLR4. To identify a possible impact of bacterial infections, as well as leaky gut issues, expression densities of TLR2 and TLR4 were determined on each leukocyte subpopulation. While not significantly altered in granulocytes (Appendix A), the expression density of TLR2 was significantly lower in SZ-derived monocytes compared to HDs (Figure 4A), and with the exception of some outliers, TLR-2 was low on lymphocytes of SZ when compared to AFs and HDs (Appendix A). Overall, TLR2 expression varied among individual AF and SZ patients (Figure 4A). TLR4 expression on monocytes was low in all but 2 patients in the AF and the HD group (Figure 4B). Therefore, ligands binding to TLR2 and -4, including antigens from bacterial infections of Gram-positive and Gram-negative bacteria, appear to be of inferior relevance in AF and SZ. To further question a possible contribution of virus infections, we included IFN-γ measurements from plasma and CSF samples (Figure 4C). In contrast to IFN-α, which was undetectable in both plasma and CSF, IFN-γ plasma concentrations were detected in a minority of SZ patients (*n* = 2/19) and about half of (*n* = 6/10) AF patients, with a median concentration of 19.43 pg/mL, whereas HD plasma levels were negative. Further, and exclusively for IFN-γ, we included measurements of plasma derived from patients with hemophagocytic lymphohistiocytosis (HLH) or macrophage activation syndrome (MAS), presenting with detectable IFN-γ concentrations in the majority (*n* = 5/7) of patients, with a median concentration of 16.64 pg/mL (Figure 4C).

**Figure 4 pharmaceuticals-15-00299-f004:**
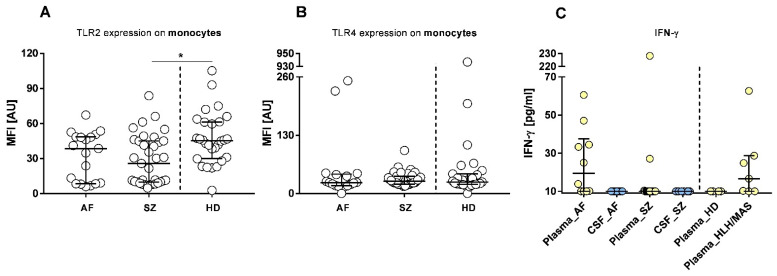
Flow cytometric profiles of pathogen receptors TLR2 and TLR4 on monocytes of patients with AF, SZ, and HDs, and IFN-γ plasma and CSF determinations of patients with AF, SZ, HDs, and patients with HLH/MAS. Monocyte expression densities of TLR2 (SZ/HDs: median: 25.79 vs. 45.35, *p* = 0.003) (**A**) and TLR4 (**B**) are shown and given as mean fluorescence intensities (MFI) or arbitrary units (AU). Plasma and CSF IFN-γ concentrations are shown (**C**). Bars indicate the medians, including interquartile ranges. Wilcoxon rank-sum test was performed to analyze differences among groups, with significant differences being marked by * *p* ≤ 0.05.

### 2.5. Lymphocyte Subpopulations Related to Virus Infections and Autoimmunity

To monitor immune activation against viruses, we performed flow cytometry of NK cell subsets in both patient groups and HDs. Relative amounts of CD56^+^ lymphocytes, CD56^+^/CD16^+^ (activated lymphocytes), and CD56^+^/CD57^+^, so-called “aged NK lymphocytes”, were non-statistically different between AF, SZ patients, and HDs (Appendix A). Nevertheless, there was a trend of higher relative amounts of CD57 co-expressing NK cells in AF when compared to SZ (Appendix A), whereas the amount of CD16^+^ (activated NK cells) was slightly higher in SZ than in AF patients, yet for both groups lower than in HDs (Appendix A). We also tested the activation of CD8^+^ cytotoxic lymphocytes in both groups which might indicate immune activation by MHC-class I restricted cytotoxic T-cells or NK effectors. In contrast to a fairly homogeneous distribution in AF patients, we found more CD25^+^/CD8^+^ coexpressing cells in individual SZ patients, but differences in the median distribution were not significant between the groups (Figure 5A).

Immune phenotypes indicating autoimmunity were expected to present with lower CD4^+^/CD25^+^ co-expressing regulatory T cells (T_reg_). Figure 5B shows that the relative amounts of CD4^+^/CD25^+^ lymphocytes in SZ scattered over a broad range: 0.59–74.75%, with the highest values in four SZ patients. The median difference in the percentage of CD25^+^/CD4^+^ lymphocytes was significant between AF and HDs (*p* = 0.05, Figure 5B). High expression densities are key characteristics to distinguish T_reg_ from activated CD4^+^ lymphocytes, thus very high CD25 positivity in CD4^+^ lymphocytes was more prevalent in SZ than in AF patients, but not significantly different from HDs (*p* = 0.005; Figure 5C). A high amount of either T_reg_ or other activated T lymphocytes would result in increased plasma concentrations of sCD25. Indeed, sCD25 plasma concentrations were lower in AF than in SZ patients (*p* = 0.03, Figure 5E). Further, median differences between SZ and HDs were highly significant (Figure 5E, *p* ≤ 0.001), and also AF patients showed higher sCD25 levels than HDs (Figure 5E). In addition to CD25, the expression of CD11b was analyzed to identify the activated lymphocyte population. As shown in Figure 5F, the median expression of CD11b was higher in AF than SZ patients and similar to HDs, but SZ patients had lower CD11b expression densities on their lymphocytes than AF and HDs. These results may hint towards a selective down modulation of CD11b^+^ lymphocytes in SZ and AF.

**Figure 5 pharmaceuticals-15-00299-f005:**
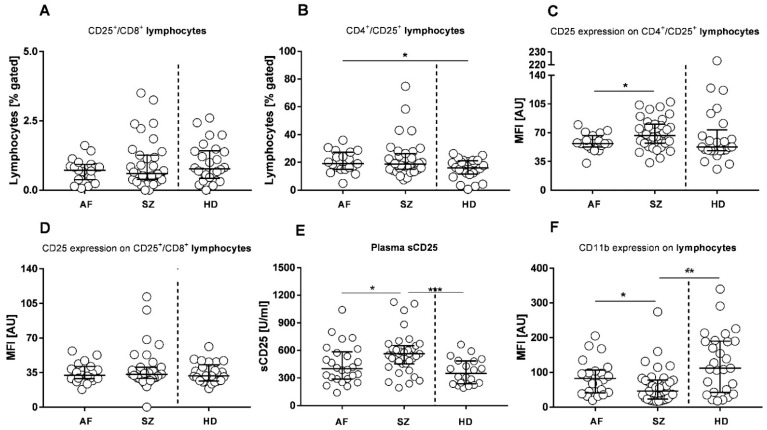
Flow cytometric profiles related to autoimmunity in patients with AF, SZ, and HDs. Relative amounts (% gated) of CD25^+^/CD8^+^ lymphocytes (**A**) and CD4^+^/CD25^+^ lymphocytes (comprising activated and T_reg_ cells, AF/HDs: median: 19.07 vs. 15.95, *p* = 0.05) (**B**) are shown. Expression densities of CD25 on T_reg_ cells (AF/SZ: median: 56.85 vs. 68.79, *p* = 0.005) (**C**), and on cytotoxic T cells (**D**) are shown and given as arbitrary units (AU). Plasma sCD25 levels are shown (AF/SZ: median: 400.0 vs. 562.0, *p* = 0.03; SZ/HDs: median: 562.0 vs. 350.1, *p* ≤ 0.001) (**E**). Lymphocyte expression densities of CD11b are shown (**F**) and given as arbitrary units (AU). Bars indicate the respective medians, including interquartile ranges. Wilcoxon rank-sum test was performed to analyze differences among groups, with significant differences being marked by * *p* ≤ 0.05, ** *p* ≤ 0.01, and *** *p* ≤ 0.001.

### 2.6. Correlation Patterns of Plasma and CSF Cytokine Levels, and Immune Phenotypes

Figure 6 shows the results of our correlation analysis for plasma- and CSF-derived cytokines with selected biomarkers determined by flow cytometry. Figure 6A,B shows that CD4^+^/CD25^+^ coexpressing lymphocyte counts correlate with sCD25 plasma concentrations in AF but not in SZ. CD11b, a marker indicating activation and competence for migration by lymphocytes, was also tested and we found strong and significant correlations with plasma levels of IL-1β, similar in both patient groups (Figure 4C,D). Further, in SZ but not AF patients, CD11b lymphocyte expression correlated significantly with plasma levels of TNF-α (Figure 4E,F). When correlating the immunosuppressive antigen CD33 on monocytes with other markers, we found a trend for a positive correlation with IL-10 plasma concentrations in AF, whereas SZ patients displayed an almost significant negative correlation (Figure 6G,H).

We then determined correlations between cytokine levels in both plasma and CSF samples. Results are given in Appendix A, respectively. In general, SZ patients presented with various high and significant correlations between plasma cytokines IL-1β, IL-8, IL-10, and TNF-α. By contrast, the correlation between plasma IL-8 and TNF-α showed a negative trend in AF. Plasma concentrations of IL-1β and TNF-α correlated with CD11b expression on lymphocytes in both patient groups. Regarding CSF concentrations, AF but not SZ patients showed a significant negative correlation between IL-8 and IL-10, indicating a stronger impact of those two cytokines in AF rather than SZ patients despite the similar systematic upregulation of IL-8 in CSF vs. plasma (Figure 1A). Accordingly, erythropoietin (EPO) correlated very strongly and significantly with IL-8 and IL-10 in a negative (IL-8) or positive (IL-10) way, respectively. In the CSF, IL-1β and S100B levels showed inverse trends for AF and SZ patients. Positive trends of S100B were restricted to IL-1β, IL-6, and IL-8, but only in AF patients. In CSF of SAH patients, S100B correlated with IL-8. Therefore, pro-inflammatory cytokines in AF patients’ CSF may result from tissue damage-derived trauma. This is further supported by strong and significant correlations between CSF levels of ferritin with IL-1β and TNF-α, respectively, in the AF but not SZ group.

## 3. Discussion

### 3.1. Inflammation in AF and SZ Patients

The etiology of psychiatric diseases, including AF and SZ, is closely associated with infections, inflammation, and/or dysregulated autoimmunity [19,20,21]. The contribution of sterile vs. infectious inflammation remains under discussion. In a relatively small cohort of AF and SZ patients, we compared immune activation parameters in both CSF and plasma to address the contribution of neurological stressors (danger-associated molecular patterns (DAMPs) vs. pathogen-derived molecular patterns (PAMPs)). Elevated IL-8 was a key pro-inflammatory marker for both AF and SZ, as previously shown [6,22]. IL-8 is a potent chemokine for neutrophils. Although neutrophils have a short lifespan in CSF, they will induce oxygen radical formation and proteolysis. A study focusing on inflammatory cytokines in Alzheimer’s disease (AD) [23] cited the work of Galimberti and colleagues [24], who showed increased CSF IL-8 levels in mild cognitive impairment as well as in AD when compared to non-demented controls. Therefore, elevated IL-8 remains a unique characteristic for neuroinflammation. While reported to not exceed CSF levels of 100 pg/mL in AD, individual patients with inflammatory neurological diseases may present with CSF IL-8 concentrations of up to 7000 pg/mL [25]. The relevance of neutrophil recruitment by IL-8 requires confirmation by differential cell counts in CSF, as suggested by a favorable methodology manuscript [26].

Both our patients’ groups presented with increased monocyte activation states, defined by elevated CD33 expression and plasma ferritin and by decreased CD163 expression on monocytes. SZ patients further displayed elevated monocyte numbers. CD33, a sialic-acid binding immunoglobulin-like lectin (Siglec-3), was elevated in monocytes of both patient groups. It interferes with monocyte and macrophage phagocytic function [27], regulates cell adhesion processes and endocytosis, and inhibits cytokine release [28]. In SZ, but not in AF patients, we found an inverse correlation of CD33 on monocytes with plasma anti-inflammatory IL-10, which further indicates inferior monocyte cell activation and function. According to Wißfeld and colleagues, CD33 also affects Aβ peptide ingestion and degradation in AD [29]. Ferritin is a biomarker secreted by M1 (pro-inflammatory) monocytes; these M1 monocytes are balanced by CD163^+^ anti-inflammatory M2 monocytes [30].

### 3.2. Different Forms of Inflammation in AF and SZ

The tissue types secreting IL-8 are likely to differ between AF and SZ patients. IL-8 may be secreted by monocytes in infection-related inflammation or produced by inflamed endothelial cells or astrocytes in the brain [31,32,33]. As reviewed, previously [34], peripheral monocytes, traveling to the brain during stress conditions, can propagate pro-inflammatory cytokines like IL-1β or IL-6 through vascular endothelial cells and hence, contribute to microglia inflammation. CD11b lymphocyte expression is significantly higher in AF than in SZ patients, and monocytes CD11b expression was tendentially elevated in AF but not in SZ patients. AF patients also displayed elevated CSF IL-1β which may originate from sterile inflammation and consecutive inflammasome activation. S100B is an established biomarker for tissue damage [35], aging [14,36], and neurological diseases, including AD [37]. Uher and colleagues correlated S100B with depressive symptoms in non-psychiatric patients [38]. Accordingly, AF patients presented with tendentially higher CSF S100B concentrations. Furthermore, S100B and CSF-derived IL-1β, IL-6, and IL-8 tendentially correlated in AF patients, which may indicate that tissue damage, S100B-, and cytokine release are functionally associated. S100B may also indicate blood–brain barrier (BBB) damage, which was likely in all our patients who presented with measurable CSF LBP (a liver biomarker) [39]. Overall, LBP was detected more frequently in CSF of our AF than in our SZ patients. This observation corresponds to increased CSF albumin concentrations of depressive disorder patients [40]. The M1 macrophage-specific ferritin also correlated with IL-1β and TNF-α, all supporting the concept of tissue damage and subsequent sterile inflammation in the AF patients’ CNS. IL-10 was also found more frequently in plasma samples of AF patients. Remarkably, low IL-8 correlated with high IL-10 in AF CSF. However, in SZ patients, IL-8 did not correspond to pro-inflammatory TNF-α and IL-1β. Long-term SZ patients are often treated by anticonvulsants with an as yet ill-defined anti-inflammatory profile, and this context needs more detailed investigation. Another option to clarify why IL-1β and TNF-α are independent of IL-8 profiles would be to screen AF and SZ patients for monocyte chemoattractant protein 1 (MCP-1), which is described to be secreted with IL-8 by inflamed macrophages [41]. While reported to be elevated together with macrophage inflammatory protein-1β (MIP-1β) in SZ [42], studies also showed unchanged MCP-1 or reduced plasma concentrations in AF patients [22,43]. In ALS, a neurodegenerative disease, elevated MCP-1 in CSF and MIP-1β were described [44]. By contrast, virus-induced encephalitis patients had low MCP-1 despite elevated IL-8 [45]. Therefore, a correlation with MCP-1 may determine IL-8’s origin in AF and SZ individuals.

In SZ patients, classical pro-inflammation appears to be less prominent. We found low CD11b expression throughout all leucocyte subpopulations but elevated CD33 monocyte expression. High CD33 expression on phagocytes not only indicates pro-inflammation in peripheral organs of SZ but may also indicate microglia activation [28]. CD33 was reported to also impair TLR-mediated cytokine activation [46] and based on the molecular interaction of Siglecs and pattern recognition receptors [47], which may explain observations by Müller and colleagues in SZ patients [48]. A third feature in SZ patients was undetectable plasma or CSF IFN-γ. Even though virus involvement in SZ has been frequently discussed [49,50], negative IFN-γ argues against virus-specific immune sensitization. The lack of IFN-γ may rather indicate T- and NK cell non-responsiveness, also supported by reduced CD11b expression. However, we found altered lymphocyte–monocyte ratios (LMRs), as previously shown for SZ patients [51]. In a large and very elegant study, Chen and colleagues further characterized SZ patients’ monocytes and found fewer nonclassically activated monocytes (CD14^+^/CD16^++^) in patients with first-episode schizophrenia [52]. However, the most remarkable change was shown in the intermediately activated monocyte population (CD14^++^/CD16^+^), which plays a major role in antigen presentation [52]. Müller and colleagues tested their hypothesis of weakened monocyte activation in the context of IL-6, finding that SZ patients’ CD33^+^ monocytes responded with higher intracellular IL-6 upon LPS stimulations than controls [53]. As a typical stress marker, IL-6 was reported to be elevated more prominently in the acute phase of depression or SZ before remission [54]. Maes and colleagues emphasized IL-6 as a biomarker for treatment-resistant schizophrenia [55]. With our cohort of long-term, treatment-resistant SZ patients [56], high IL-6 was detected only in individual patients and was associated with impaired antigen presentation and intermediately activated monocytes. Murphy and colleagues also described a paradox state of weaker NF-κB activation in SZ patients, yet with high pro-inflammatory cytokine concentrations in the brain [57]. Corsi-Zuelli and Deakin hypothesized this “microglia paradox” to be, in part, caused by dysregulated T_reg_ cells in the brain [58]. The consequent comment by Murphy and Weickert emphasizes the “T_reg_-astrocyte-microglia theory” for further discussion [59]. In addition, our SZ patients displayed high numbers of immunosuppressive CD123^+^ pDCs. In conclusion, some SZ patients provide evidence for autoimmune dysfunction, hypothetically based on pDC activation by tonic TLR stimulation by as yet undefined ligands [60].

By contrast, immune phenotypes of AF patients displayed elevated CD11b expression in all leukocyte subtypes, indicating high affinity to endothelial ICAM-1. Of the CD11b^+^ T cells in AF patients, higher numbers may be T helper 1 (T_H1_) or T_r1_ rather than T_reg_ cells [61,62], which may also explain higher IFN-γ concentrations in AF. A possible inducer hereby is IL-27, which may suppress T_reg_ phenotype generation, also on the CD25 level [63]. Indeed, we found higher CD25 expression densities on CD4^+^/CD25^+^ lymphocytes in SZ. These CD4^+^/CD25^+^ cells may correspond to so-called hypofunctional T_reg_ (h-T_reg_), coined by Corsi-Zuelli and colleagues [64]. We fully agree with the authors that better T_reg_ subclassification is necessary. The present study may support the autoimmune phenotype in SZ despite high CD4^+^/CD25^+^ T cell counts. Systematically increased sCD25 may indicate higher protease activity [65], and low regulatory capacity of CD4^+^/CD25^+^ T_reg_ cells, thus further supporting autoimmunity [66]. A positive correlation of CD4^+^/CD25^+^ T cells and sCD25 plasma concentrations may clearly indicate autoimmune dysregulation in individual SZ patients. In CSF, a proper description of T_reg_ (sub)sets would offer a promising topic to extend previous studies [67].

Extension and refinement of the here presented immune phenotypes should also include therapeutic aspects for individual patients, as suggested by other studies [68]. A theranostic approach would also include long-term psychotic treatment. As an example, clozapine is known to influence TLR3-specific inflammasome activation [69], and modifications of pro-inflammatory pathways have also been described for other psychotic drugs [70,71]. Consequently, we favor multiple follow-up measurements of cytokines associated with successful or non-successful treatment in affective and schizophrenic disorders. This manuscript describes a selective and new marker profile that could be applied.

One of the major drawbacks of this explorative study is its small size and relative heterogenous patients’ characteristics. However, we found evidence for both, systemic and brain-derived inflammation in long-term and treatment-resistant AF and SZ patients. The common pro-inflammatory phenotype consisted of M1 macrophage activation in the periphery and high oxidative stress in the brain, as indicated by elevated IL-8. AF patients were further characterized by T_H1_/T_r1_ cell activation and measurable IFN-γ. Elevated CSF IL-1β supports the hypothesis of DAMP-induced inflammasome activation in the brain, with infiltrating lymphocytes and/or monocytes likely being the cause of tissue damage and oxidative stress induced by IL-8. In SZ, functionally impaired monocytes and CD4^+^/CD25^+^ lymphocytes, high sCD25, and elevated pDCs and CD33^+^ monocytes may lead to impaired cooperation between APC and T and B lymphocytes and thus promote the manifestation of an autoimmune phenotype. We are looking forward to larger studies making use of the current findings.

## 4. Materials and Methods

### 4.1. Patient Cohorts

The current study was approved by the Ethics Committee of Ulm University (No. 17-04/2006, 98/2019, 400/20). A total of *n* = 68 patients were included, successively recruited at the Department Psychiatry and Psychotherapy II, Guenzburg, Ulm University. Inclusion criteria were schizophrenic (SZ) or affective (AF) spectrum disorder (ICD-10 F20-F25 and F30-F33), and informed written consent. All patients received multiple treatments without long-term improvement. Exclusion criteria: Increased bleeding risk, increased cerebral pressure (excluded by brain imaging), fever, leukopenia, suspected meningoencephalitis, or multiple sclerosis (MS). AF patients (*n* = 29) were older (mean: 41 years) than SZ patients (*n* = 39) (mean: 35 years). A lifetime ICD-10 psychiatric diagnosis was made by clinical consensus including extended laboratory analysis, MRI and/or CCT, EEG, ECG. By OPCRIT checklist ICD-10 versus DSM-IV diagnosis was compared. Schizophrenia spectrum included schizoaffective and affective spectrum included bipolar disorder. Patients with undetermined clinical aspects, e.g., possible primary neurological disorder or questionable medical disorder, underwent an interdisciplinary case conference with participating specialists, which often prompted specific investigations. Exclusively for cerebrospinal fluid (CSF), a total of *n* = 5 patients with subarachnoid hemorrhage (SAH) were recruited (Ethics Approval no. 82/07, Ulm University) as controls for non-psychiatric neurotrauma. As a reference group for plasma, samples from *n* = 7 patients with hemophagocytic lymphohistiocytosis (HLH) or macrophage activation syndrome (MAS) were included, based on routine analysis in the context of clinical examination. Further patient information regarding age and sex is provided in Table 2.

### 4.2. Healthy Donors

As healthy controls, a total of *n* = 44 healthy volunteers were included. Their respective immune phenotypes and plasma biomarkers were also based on routine analysis in the context of routine clinical examination. Results of all healthy donors were further pseudonymized before processing for statistical analysis. Further information regarding age and sex is provided in Table 2.

### 4.3. Cytokine Measurements

Cytokine measurements for interleukins (IL)-6, -8, -10, -1β, tumor necrosis factor (TNF-)-α, soluble (s)CD25, lipopolysaccharide binding protein (LBP), erythropoietin (EPO), and ferritin were performed using a chemiluminescence-based, validated assay (Immulite^®^1000, www.siemens-healthineers.com, accessed on 23 February 2022) according to the manufacturer’s protocol. Interferons (IFN-) -α, -γ, and S100 calcium-binding protein B (S100B) were quantified by commercially available ELISAs (www.bio-techne.com, www.abnova.com, respectively, both accessed on 23 February 2022) according to the manufacturer’s protocols.

### 4.4. Flow Cytometry

The expression of surface antigens on blood leukocytes was performed by using fluorescently labeled antibodies directed against cluster of differentiation (CD) antigens. Analysis was performed using the FACSCalibur™ device (www.bdbiosciences.com, accessed on 23 February 2022) and corresponding software (BD CellQuest^TM^ Pro, version 6.1, www.bdbiosciences.com, accessed on 23 February 2022). Antibodies recognizing CD2, CD4, CD8, CD11b, CD14, CD16, CD19, CD25, CD33 (Siglec-3), CD45, CD56, CD57, CD64, CD66b, CD123, CD163, CD282 (TLR2), CD284 (TLR4), HLA-DR, and the isotype control were purchased from BD Biosciences (www.bdbiosciences.com, accessed on 23 February 2022).

### 4.5. Statistics

All data visualization and statistical analyses were performed using the GraphPad PRISM software (version 9.1.1, www.graphpad.com, accessed on 23 February 2022).

## Figures and Tables

**Figure 1 pharmaceuticals-15-00299-f001:**
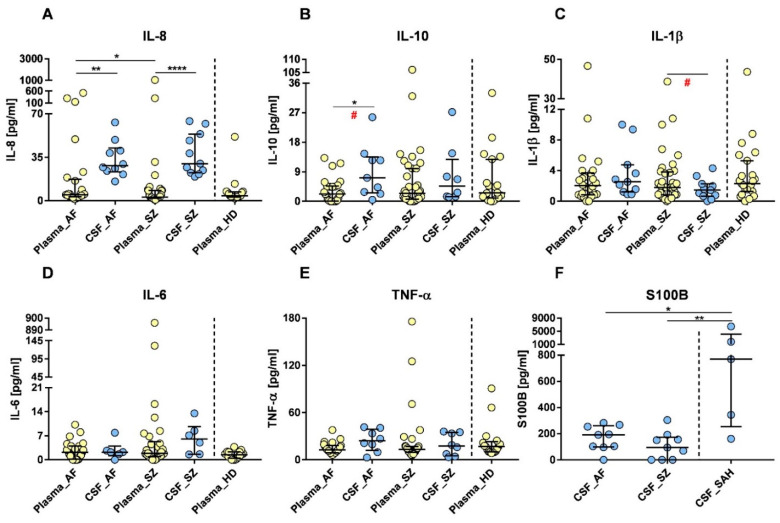
Biomarker concentrations in plasma and cerebrospinal fluid (CSF) of patients with affective (AF), and schizophrenic (SZ) spectrum disorders, and healthy donors (HDs). Concentrations of Interleukins (IL-)-8 (**A**), IL-10 (**B**), IL-1β (**C**), IL-6 (**D**), tumor necrosis factor α (TNF-α) (**E**), and calcium-binding protein B (S100B) (**F**, CSF only) are shown as scatter plots, with bars indicating the medians, including interquartile ranges. Wilcoxon rank-sum test was performed to analyze differences among groups, with significant differences being marked by * *p* ≤ 0.05, ** *p* ≤ 0.01, and **** *p* ≤ 0.0001. Wilcoxon matched-pairs signed-rank test was performed to analyze differences between patient-matched pairs (plasma vs. CSF), with significant differences being marked by # *p* ≤ 0.05. AF patients showed elevated IL-8 concentrations compared to the SZ group (median: 4.8 pg/mL vs. 2.95 pg/mL, *p* = 0.03, Figure 1A). CSF levels of IL-8, compared to the respective plasma concentrations, were systematically elevated in both AF (median: 28.20 pg/mL vs. 4.8 pg/mL, *p* = 0.002, Figure 1A) and SZ patients (median: 29.90 pg/mL vs. 2.95 pg/mL, *p* < 0.0001, Figure 1A). For IL-10, a significant difference regarding plasma vs. CSF was observed in the AF group (median: 2.20 pg/mL vs. 7.14 pg/mL, *p* = 0.02, Figure 1B). IL-1β in CSF was similar in AF and SZ (median: 2.52 pg/mL vs. 2.03 pg/mL, *p* = 0.40, Figure 1C). In the SZ group, and when compared for matched samples, IL-1β was significantly lower in CSF compared to plasma (median: 1.45 pg/mL vs. 1.80 pg/mL, *p* = 0.02, Figure 1C).

**Figure 2 pharmaceuticals-15-00299-f002:**
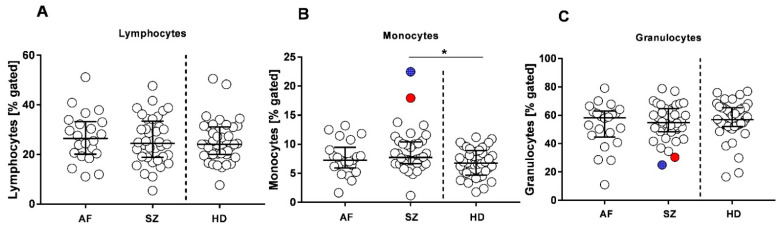
Relative amounts of leukocyte subpopulations in patients with AF, SZ, and HDs. Lymphocyte (**A**), monocyte (**B**), and neutrophilic granulocyte (**C**) amounts are shown from each patient (% gated). Bars indicate the respective medians, including interquartile ranges. Two outlier patients in the SZ group are color labeled (red and blue symbols, respectively). Wilcoxon rank-sum test was performed to analyze differences among groups, with significant differences being marked by * *p* ≤ 0.05.

**Figure 6 pharmaceuticals-15-00299-f006:**
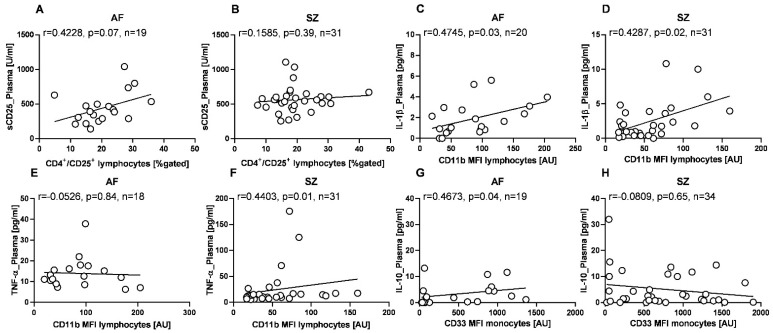
Correlation patterns for selected cytokine concentrations and immune phenotypes, in patients with AF and SZ. Shown correlations are CD4^+^/CD25^+^ lymphocyte counts vs. sCD25 plasma concentrations for AF and SZ (**A**,**B**, respectively), CD11b lymphocyte expression densities vs. IL-1β plasma concentrations for AF and SZ (**C**,**D**, respectively), CD11b lymphocyte expression densities vs. TNF-α plasma concentrations for AF and SZ (**E**,**F**, respectively), and CD33 monocyte expression densities vs. IL-10 plasma concentrations for AF and SZ (**G**,**H**, respectively). Expression densities are given as arbitrary units (AU), and cell counts are given as %-gated cells. Data are presented as scatter plots, with the line representing the linear regression fit, and r indicating the Spearman correlation coefficient. Shown correlations present relations in the majority of the investigated cohort, yet in some figures, individual outlier values were excluded from the correlation analysis.

**Table 1 pharmaceuticals-15-00299-t001:** Neutrophil–lymphocyte ratios (NLR) and lymphocyte–monocyte ratios (LMR) of patients with affective (AF) and schizophrenic (SZ) spectrum disorders, and healthy donors (HDs). Values are given as medians, including ranges. Wilcoxon rank-sum test was performed to analyze differences among groups, with significant differences being marked by * *p* ≤ 0.05.

	Neutrophil–Lymphocyte Ratio (NLR) [Median]	Lymphocyte–Monocyte Ratio (LMR) [Median]
AF	2.06 (range 0.56–6.62)	3.37 (range 1.71–6.80)
SZ	2.34 (range 0.64–14.57)	3.07 (range 1.19–9.01) (* vs. HDs, *p* = 0.02)
HDs	2.44 (range 0.33–4.93)	4.06 (range 1.44–10.23)

**Table 2 pharmaceuticals-15-00299-t002:** Cohort characteristics of patients and controls.

Group	ICD-10 Class	Total	Male	Female	Mean Age
Affective spectrum disorder (AF)	F30-F33	*n* = 29	*n* = 13	*n* = 16	41 (range 28–69)
Schizophrenic spectrum disorder (SZ)	F20-F25	*n* = 39	*n* = 22	*n* = 17	35 (range 20–65)
Subarachnoid hemorrhage (SAH)	I60	*n* = 5	*n* = 2	*n* = 3	51 (range 40–68)
Healthy donors (HDs) ^1^	-	*n* = 44	*n* = 19	*n* = 23	34 (range 18–68)
HLH/MAS	D76.1	*n* = 7	*n* = 3	*n* = 4	30 (range 15–44)

^1^ Of the healthy donor group, *n* = 2 individuals were completely anonymized. Therefore, information on age or sex cannot be provided.

## Data Availability

Data is contained within the article and Appendix A.

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
