# Peer review of "Old and New Biomarkers for Infection, Inflammation, and Autoimmunity in Treatment-Resistant Affective and Schizophrenic Spectrum Disorders"

_pharmaceuticals, 2022, doi:10.3390/ph15030299_

Round 1

Reviewer 1 Report

The authors have submitted a manuscript of investigating whether some biomarkers such as IL-6, -8, -10, -1beta, TNF-alpha, soluble CD25, LBP, EPO, ferritin, IFN-alpha, -gamma, S100B, and cell surface antigens such as CDs were quantified in plasma, CSF, or monocytes from patients with affective disorders and schizophrenia, as compared with healthy control subjects. Based on the results, an increase in the levels of IL-8 in was found significant in patients with affective disorders and schizophrenia, suggesting that the pro-inflammatory factor IL-8 could be a biomarker for the affective disorders and schizophrenia. The main finding of the present study regarding the IL-8 hypothesis is conceivably the novel finding and is attractive. This story would be their strength because of some novel findings. The issue has widely been focused, and, overall, the impact of this study appears high, and the information provided may offer something substantial that helps advance our understanding of novel biomarkers against human brain disorders.

The authors should show or at least describe with additional reference the critical difference of the observed IL-8 increments between affective disorders/schizophrenia and the other brain disorders. Please explain authors’ perspective of this points in the discussion section of the revised manuscript, because the measures of IL-8 would be identical or indistinguishable between affective disorders/schizophrenia and the other brain disorders, based on the results.

Author Response

Point 1: The authors should show or at least describe with additional reference the critical difference of the observed IL-8 increments between affective disorders/schizophrenia and the other brain disorders. Please explain authors’ perspective of this points in the discussion section of the revised manuscript, because the measures of IL-8 would be identical or indistinguishable between affective disorders/schizophrenia and the other brain disorders, based on the results.

Response 1: Thank you very much for this important argument which may contribute to our understanding of IL-8 pathology in AF and SZ by identification of the cell type secreting IL-8. For the revised manuscript, we inserted into the discussion some additional aspects regarding the origin of IL-8 in CSF. Some correlative trends may reveal this cytokine to be the result of cell infiltration and subsequent endothelial damage in AF patients, however, the origin of IL-8 in SZ patients has yet to be elucidated. Determining MCP-1 as an important monocyte-attracting chemokine may solve this issue (Proma MA, Daria S, Nahar Z, Ashraful Islam SM, Bhuiyan MA, Islam MR. Monocyte chemoattractant protein-1 levels are associated with major depressive disorder. J Basic Clin Physiol Pharmacol. 2022 Jan 5. doi: 10.1515/jbcpp-2021-0132. Epub ahead of print. PMID: 34983131). Please also see the modified supplementary Figure S1, and Table S2, where we determined the IL-8 levels for our SAH samples, and included them into our correlation analysis, respectively.

Reviewer 2 Report

The authors investigated immune profiles in patients with affective and schizophrenic spectrum disorder.

Here are some questions for consideration related to this study:

Minor issues:

My general impression is that authors overstated their results.  The interpretation is „ambitious“ considering the number of samples investigated.

The authors should specify forms of schizophrenic and affective spectrum disorders observed in patients that donated samples. If patients were of diverse forms (particularly for affective spectrum disorder), the generalization of the obtained data is questionable and should be commented by the authors.

In general, the text is very intense and not easy to follow.

Conclusions (hypothesis) that affective spectrum disorders are related to viral infections and that there is connection between schizophrenia and autoimmunity are not new.

Nevertheless, Figure 3C - IFN-γ plasma concentrations were detected in a minority of SZ patients (n=2/19) and about half of (n=6/10) AF patients. Due to the small number of patients with the observable INF- γ , and without HD values, any conclusion about the contribution of viral infection is of inferior relevance.

Are the investigators sure that outliers are not due to some methodological issue? For example, levels of IL-8 in plasma of SC patients are below 35, the majority of the samples is below 15 pg/ml, and one sample is around 1000.

Many of the findings have been published by other authors (bellow is the list which is not final). Please explain in details the novelty and contribution (as well as importance) of the obtained data to the field  and clinics.

IL-1β in CSF od SZ - https://www.nature.com/articles/mp200952

Levels of cytokines and immune markers in SZ  https://www.ncbi.nlm.nih.gov/pmc/articles/PMC5449450/

https://pubmed.ncbi.nlm.nih.gov/29239785/

https://www.ncbi.nlm.nih.gov/pmc/articles/PMC6365449/

https://jneuroinflammation.biomedcentral.com/articles/10.1186/s12974-018-1197-2/figures/1

https://pubmed.ncbi.nlm.nih.gov/25022963/

Regulatory T cells in SC - https://pubmed.ncbi.nlm.nih.gov/32221694/

Expression of CD163+ in SZ - https://www.nature.com/articles/s41380-018-0235-x

Number of blood cells in SZ - https://pubmed.ncbi.nlm.nih.gov/32178747/

https://www.psychiatria-danubina.com/UserDocsImages/pdf/dnb_vol28_sup2/dnb_vol28_sup2_32.pdf

Affective disorder

https://www.ncbi.nlm.nih.gov/pmc/articles/PMC7056473/

https://www.frontiersin.org/articles/10.3389/fpsyt.2019.00030/full

https://www.ncbi.nlm.nih.gov/pmc/articles/PMC8474635/

Figure 1 – comparison with the control group should be performed

Minor issues:

Abstract is repetitive

Line 72 – IL-10 is not a pro-inflammatory marker

Line 79 – „However, inflammasome-related IL-1β is more prevalent in the peripheral blood“ – not correct, only for SZ group

All lines indicating median values are not of the same thick

Line 170-173 – „While not significantly altered in granulocytes, expression densities of TLR2 were different between  groups in both lymphocytes and neutrophilic granulocytes, when compared with HDs.“ – unclear, this part of the text relates to what exactly?

Line 173 - Figure 4 shows impressive heterogeneity in AF and SZ, despite the fact that TLR2 was  lower in AF and SZ when compared to healthy controls (Figure 4 A). – monocytes are indicated on 4A, only SZ is lower than HD

HD is missing in Figure 4C

Please comment why all the samples were not included in correlational patterns (Figure 6)

Figure legend of Figure 6 should be corrected

Table 2 – numbers for AF do not match

Line 193 – „Nevertheless, there was a trend of higher relative amounts of CD57 co-expressing NK cells in AF when compared to SZ, and the amount of CD16+, activated NK cells was slightly higher than in AF patients.“ -please correct the sentence

Line 211 – „...however, both patient groups had higher sCD25 plasma concentrations than HDs“ – not labelled on Figure 5E

Line 216 – „These results may hint to selective down modulation of CD11b+  lymphocytes in AF“ – there is no statistical significance, overstatement

Line 292 – „In both patient groups, we also found evidence for innate immune activation and 292 inflammation by increased numbers and activation states of monocytes and macrophages.“ – should be explained in more details, number of monocytes was not increased (Figure 2B)

Line 353 – LMRs – for clarity should be written as leukocyte/monocyte ratio

Author Response

Point 1: My general impression is that authors overstated their results.  The interpretation is „ambitious“ considering the number of samples investigated.

Response 1: This reviewer’s argument is well taken. We have detailed immunological and clinical information on each individual patient presented, allowing pro- and anti-inflammatory immunity, autoimmunity, as well as bacterial- and virus-specific immune activation at the given time point presented. Some of this detailed information is included in the revised version of the manuscript.

Point 2: The authors should specify forms of schizophrenic and affective spectrum disorders observed in patients that donated samples. If patients were of diverse forms (particularly for affective spectrum disorder), the generalization of the obtained data is questionable and should be commented by the authors.

Response 2: The patients’ cohort is unique by therapy-resistant pathology (all in inpatient treatment protocols) of the affective and schizophrenic spectrum as described in a previous study focused on routine high standard CSF analytics (Bechter K, Reiber H, Herzog S, Fuchs D, Tumani H, Maxeiner HG. Cerebrospinal fluid analysis in affective and schizophrenic spectrum disorders: identification of subgroups with immune responses and blood-CSF barrier dysfunction. J Psychiatr Res. 2010 Apr;44(5):321-30. doi: 10.1016/j.jpsychires.2009.08.008. Epub 2009 Sep 30. PMID: 19796773.). We are well aware that typical and atypical anti-psychotic drugs, neuroleptics, and anti-inflammatory agents may well affect inflammation determined on the level of cytokines and soluble receptors. In previous studies, cell count and albumin ratios have been described as measures for inflammation, which are not different between first episode and multi-episode cases (Rattay TW, Martin P, Vittore D, Hengel H, Cebi I, Tünnerhoff J, Stefanou MI, Hoffmann JF, von der Ehe K, Klaus J, Vonderschmitt J, Herrmann ML, Bombach P, Al Barazi H, Zeltner L, Richter J, Hesse K, Eckstein KN, Klingberg S, Wildgruber D. Cerebrospinal fluid findings in patients with psychotic symptoms-a retrospective analysis. Sci Rep. 2021 Mar 30;11(1):7169. doi: 10.1038/s41598-021-86170-w. PMID: 33785807; PMCID: PMC8010098.). In summary, we favor multiple follow-up measurements of cytokines related to successful or non-successful treatment in affective and schizophrenic disorders. This manuscript describes a selective and new marker profile in order to stratify biomarker profiles on individual patient levels, which could be applied in future studies.

Point 3: In general, the text is very intense and not easy to follow.

Response 3: The text has been improved accordingly.

Point 4: Conclusions (hypothesis) that affective spectrum disorders are related to viral infections and that there is connection between schizophrenia and autoimmunity are not new.

Response 4: The reviewer is fully correct, however, this manuscript provides new biomarkers to identify the viral (sCD25) and autoimmune contribution (CD4+/CD25+ lymphocytes), etc. in a quantitative way for an individual patient at several time points.

Point 5: Nevertheless, Figure 3C - IFN-γ plasma concentrations were detected in a minority of SZ patients (n=2/19) and about half of (n=6/10) AF patients. Due to the small number of patients with the observable INF- γ , and without HD values, any conclusion about the contribution of viral infection is of inferior relevance.

Response 5: The reviewer is correct in that Fig. 4 C shows that only few SZ patients display increased interferon-gamma, despite a likely role of virus infections. However, in the absence of activated NK‑ and cytotoxic lymphocytes (CTLs), an anergic antivirus response is likely. In the revised version of the manuscript, we included another data set of patients with virus-induced neuropathology, and concommitatly increased NK cells and CTLs. Further, HD values for IFN-γ are now also displayed in the revised Fig. 4 C.

Point 6: Are the investigators sure that outliers are not due to some methodological issue? For example, levels of IL-8 in plasma of SC patients are below 35, the majority of the samples is below 15 pg/ml, and one sample is around 1000.

Response 6: Thank you for this important argument. We now checked back each individual outlier of high or low inflammatory cytokines. Accordingly, multiple treatment options may affect pro-inflammatory cytokines such as IL-1, IL-6, and TNF-α. However, IL-8 as a biomarker does not appear to be affected. As an example, clozapine has been demonstrated to influence TLR3-specific inflammasome activation (Giridharan VV, Scaini G, Colpo GD, Doifode T, Pinjari OF, Teixeira AL, Petronilho F, Macêdo D, Quevedo J, Barichello T. Clozapine Prevents Poly (I:C) Induced Inflammation by Modulating NLRP3 Pathway in Microglial Cells. Cells. 2020 Feb 28;9(3):577. doi: 10.3390/cells9030577. PMID: 32121312; PMCID: PMC7140445.). In future studies the authors intend to follow pharmacological effects in greater detail. According to results presented in supplementary Table S1, pro-inflammatory cytokines correlate in SZ but not in AF patients, indicating the influence by therapy. In a study with COVID-19 patients with inflammatory neurological diseases CSF IL‑8 may reach up to 7000 pg/ml, even though the majority did not exceed 50 pg/ml (Espíndola OM, Gomes YCP, Brandão CO, Torres RC, Siqueira M, Soares CN, Lima MASD, Leite ACCB, Venturotti CO, Carvalho AJC, Torezani G, Araujo AQC, Silva MTT. Inflammatory Cytokine Patterns Associated with Neurological Diseases in Coronavirus Disease 2019. Ann Neurol. 2021 May;89(5):1041-1045. doi: 10.1002/ana.26041. Epub 2021 Feb 24. PMID: 33547819; PMCID: PMC8014707.).

Point 7: Many of the findings have been published by other authors (bellow is the list which is not final). Please explain in details the novelty and contribution (as well as importance) of the obtained data to the field  and clinics.

Response 7: The reviewer is fully correct. The results of our small treatment-resistant cohort support previous knowledge in AF and SZ. As a novelty, we provide evidence for sCD25 as a biomarker from CD25+ lymphocytes (Treg) to qualify and quantify autoimmune activation in SZ. Soluble CD25 can be measured by a validated ELISA, and is highyly suitable for follow-up studies. Soluble CD25 is a result from protease activity (Damoiseaux J. The IL-2 - IL-2 receptor pathway in health and disease: The role of the soluble IL-2 receptor. Clin Immunol. 2020 Sep;218:108515. doi: 10.1016/j.clim.2020.108515. Epub 2020 Jul 1. PMID: 32619646.) conditioning inactivation of Treg. Further, CD33 turned out to identify impaired phagocytic function in SZ, but requires flow cytometry. Finally, low CD11b expression on lymphocytes may serve as another biomarker for treatment-resistant SZ.

The reviewer’s suggestions imply the future relevance of theranostic approaches in SZ. These arguments have been included in the revised version of the manuscript.

IL-1β in CSF od SZ - https://www.nature.com/articles/mp200952

Levels of cytokines and immune markers in SZ  https://www.ncbi.nlm.nih.gov/pmc/articles/PMC5449450/

https://pubmed.ncbi.nlm.nih.gov/29239785/

https://www.ncbi.nlm.nih.gov/pmc/articles/PMC6365449/

https://jneuroinflammation.biomedcentral.com/articles/10.1186/s12974-018-1197-2/figures/1

https://pubmed.ncbi.nlm.nih.gov/25022963/

Regulatory T cells in SC - https://pubmed.ncbi.nlm.nih.gov/32221694/

Expression of CD163+ in SZ - https://www.nature.com/articles/s41380-018-0235-x

Number of blood cells in SZ - https://pubmed.ncbi.nlm.nih.gov/32178747/

https://www.psychiatria-danubina.com/UserDocsImages/pdf/dnb_vol28_sup2/dnb_vol28_sup2_32.pdf

Affective disorder

https://www.ncbi.nlm.nih.gov/pmc/articles/PMC7056473/

https://www.frontiersin.org/articles/10.3389/fpsyt.2019.00030/full

https://www.ncbi.nlm.nih.gov/pmc/articles/PMC8474635/

Point 8: Figure 1 – comparison with the control group should be performed

Response 8: None of the cytokines analyzed in Figure 1 were significantly different from the control group (HD). This is now further emphasized in the revised version of the manuscript.

Lines 91-93 now reads: For IL-8, IL-10, IL-1β, IL-6, and TNF-α, plasma concentrations of both patients’ groups were not significantly different from HDs.

Minor issues:

Point 9: Abstract is repetitive

Response 9: The reviewer is fully correct. We now improved the abstract accordingly.

Point 10: Line 72 – IL-10 is not a pro-inflammatory marker

Response 10: We now specified the properties of IL-10:

Lines 62-63 now reads: Peripheral blood plasma and cerebrospinal fluid (CSF) were tested for pro-inflammatory biomarkers IL-8 and IL-1β, as well as the anti-inflammatory marker IL-10.

Point 11: Line 79 – „However, inflammasome-related IL-1β is more prevalent in the peripheral blood“ – not correct, only for SZ group

Response 11: This context is now better specified in the revised version of the manuscript.

Lines 72-73 now reads: However, despite similar plasma levels, inflammasome-related IL-1β is more prevalent in the peripheral blood of SZ rather than AF patients.

Point 12: All lines indicating median values are not of the same thick

Response 12: In the revised manuscript, lines indicating median values are now all of the same thickness.

Point 13: Line 170-173 – „While not significantly altered in granulocytes, expression densities of TLR2 were different between  groups in both lymphocytes and neutrophilic granulocytes, when compared with HDs.“ – unclear, this part of the text relates to what exactly?

Response 13: Thank you very much for addressing this mistake, which has been corrected accordingly.

Lines 166-172 now reads: While not significantly altered in granulocytes (Figure S2 H), expression density of TLR2 was significantly lower in SZ-derived monocytes compared to HDs (Figure 4 A), and with the exception of some outliers, trendwise low on lymphocytes when compared to AFs and HDs (Figure S2 G). Figure 4 A further shows impressive heterogeneity in TLR2 expression for AF and SZ. For the AF, however, a trend of lower TLR2 expression, compared to HDs, could also be observed.

Point 14: Line 173 - Figure 4 shows impressive heterogeneity in AF and SZ, despite the fact that TLR2 was  lower in AF and SZ when compared to healthy controls (Figure 4 A). – monocytes are indicated on 4A, only SZ is lower than HD

Response 14: This context now has been corrected. Cf. Response 13.

Point 15: HD is missing in Figure 4C

Response 15: For comparison, the revised Figure 4 C now includes values for HDs, and patients with Hemophagocytic lymphohistiocytosis (HLH) or macrophage activation syndrome (MAS).

Lines 180-184 now reads: , whereas HD plasma levels were negative. Further, and exclusively for IFN‑γ, we included measurements of plasma derived from patients with Hemophagocytic lymphohistiocytosis (HLH) or macrophage activation syndrome (MAS), presenting with detectable IFN-γ concentrations in the majority (n=5/7) of patients, with a median concentration of 16.64 pg/ml (Figure 4 C).

Point 16: Please comment why all the samples were not included in correlational patterns (Figure 6)

Response 16: Figure 6 focuses solely on trendwise significant correlations regarding sCD25, CD11b, and CD33 in order to emphasize these antigens in the light of new biomarkers for AF and SZ. We carefully checked clinical conditions of individual patients at the time of analysis, and decided to exclude 2-3 data points for reasons of inflammation/infection possible unrelated to the underlying psychiatric disease. All other correlations addressing cytokine concentrations can be found in Tables S1 and S2, respectively.

Point 17: Figure legend of Figure 6 should be corrected

Response 17: The figure legend in the revised Figure 6 has been corrected.

Point 18: Table 2 – numbers for AF do not match

Response 18: The numbers have been corrected for the revised manuscript.

Point 19: Line 193 – „Nevertheless, there was a trend of higher relative amounts of CD57 co-expressing NK cells in AF when compared to SZ, and the amount of CD16+, activated NK cells was slightly higher than in AF patients.“ -please correct the sentence

Response 19: The difference between AF, SZ, and HDs regarding their NK cell subsets is now better described, also referencing supplemnetary material.

Lines 196-200 now reads: HDs (Figure S2 A‑C). Nevertheless, there was a trend of higher relative amounts of CD57 co-expressing NK cells in AF when compared to SZ (Figure S2 C), whereas the amount of CD16+ (activated NK cells) was slightly higher in SZ than in AF patients, yet for both groups lower than in HDs (Figure S2 B).

Point 20: Line 211 – „...however, both patient groups had higher sCD25 plasma concentrations than HDs“ – not labelled on Figure 5E

Response 20: The reviewer is correct since the range of sCD25 is not different between AF and HDs.

Lines 215-217 now reads: Further, median differences between SZ and HDs were highly significant (Figure 5E, p0.001), and also AF patients showed a tendency of higher sCD25 levels than HDs (Figure 5 E)

Point 21: Line 216 – „These results may hint to selective down modulation of CD11b+  lymphocytes in AF“ – there is no statistical significance, overstatement

Response 21: The revised version of the manuscript now describes that only SZ had a significantly lower expression of CD11b on lymphocytes, compared to AF and HDs. For AF, the the CD11b expression on lymphocytes was only trendwise lower than in HDs, yet not significantly different.

Lines 218-222 now reads: As shown in Figure 5 F, the median expression of CD11b was higher in AF than SZ patients, and similar to HDs, but SZ patients had lower CD11b expression densities on their lymphocytes than AF and HDs. These results may hint towards a selective down modulation of CD11b+ lymphocytes in SZ and trendwise also in AF patients.

Point 22: Line 292 – „In both patient groups, we also found evidence for innate immune activation and 292 inflammation by increased numbers and activation states of monocytes and macrophages.“ – should be explained in more details, number of monocytes was not increased (Figure 2B)

Response 22: In the revised version of the manuscript, this part of the discussion has been improved accordingly. Further, it has been explained that only SZ presented with elevated monocyte numbers.

Lines 296-299 now reads: We also found evidence for inflammation and innate immune activation. In detail, both patient groups presented increased monocyte activation states, defined by elevated monocytic expression of CD33, and further, elevated plasma ferritin and decreased CD163+ monocytes. SZ patients further showed elevated monocyte numbers.  

Point 23: Line 353 – LMRs – for clarity should be written as leukocyte/monocyte ratio

Response 23: Done.

Round 2

Reviewer 1 Report

The updated manuscript has been revised adequately according to the reviewers' comments including me. I have no more comments, and now the paper is recommended to be accepted for publication in the journal Pharmaceuticals.

Author Response

Many thanks to reviewer 1 for supportive arguments

Reviewer 2 Report

The authors have improved the manuscript, although further improvements are needed before acceptance. The Discussion section could be shortened, the text is still hard to follow, and English editing is suggested.

Here are some minor issues that should be corrected.

Line 16 (abstract) - Plasma-derived biomarkers were ferritin, elevated in AF and SZ.- please correct the sentence

Line 141 – briefly define iCAM-1

Line 240 - revealed a, yet not significant – should be corrected

Line 254 – „IL-1β and TNF-α, both - regarding their plasma concentrations - highly correlating with CD11b expression on lymphocytes in both patient  groups, showed a positive correlation.“ - unclear

Plasma concentrations of IL-1β and TNF-α positively correlated with CD11b expression on lymphocytes in both patient groups.

Line 260 - correlated very strongly and significantly

Line 264 – were restricted

Line 265 - This shifts AF patients more towards high levels of pro-inflammatory cytokines in the brain being caused by tissue damage-derived trauma. – please correct English

Line 358 –„ with an as yet ill-defined anti-inflammatory profile“ – please correct English

Line 364 - a neurodamage disease – please correct English
